Assessing the quantitative relationships between the impervious surface area and surface heat island effect during urban expansion

Ma Xiaoliang 1 2
Peng Shuangyun 1 2 pengshuangyun@ynnu.edu.cn
1 Faculty of Geography, Yunnan Normal University , Kunming, Yunnan , China
2 Center for Geospatial Information Engineering and Technology of Yunnan Province , Kunming, Yunnan , China
Ghosh Tilottama
Electronic publication date: 2021 Jul 23
Publication date: 2021
Volume: 9
Electronic Location ID: e11854
Received 2020 Dec 3; Accepted 2021 Jul 4
Copyright: © 2021 Ma and Peng
Copyright year: 2021
Copyright holder: Ma and Peng
License: This is an open access article distributed under the terms of the Creative Commons Attribution License, which permits unrestricted use, distribution, reproduction and adaptation in any medium and for any purpose provided that it is properly attributed. For attribution, the original author(s), title, publication source (PeerJ) and either DOI or URL of the article must be cited.
License URL: https://creativecommons.org/licenses/by/4.0/

Keywords: Urban expansion, Surface urban heat island, Tempo-spatial variation, Geospatial analysis, Plateau lakeside city

Funding: National Natural Science Foundation of China 41971369, 41861051, 41561086, 41661082 and 41461038 This research was financially supported by the National Natural Science Foundation of China (No. 41971369, 41861051, 41561086, 41661082, 41461038). The funders had no role in study design, data collection and analysis, decision to publish, or preparation of the manuscript.

==============================
As an important component of underlying urban surfaces, the distribution pattern and density of the impervious surface area (ISA) play an important role in the generation of surface urban heat island (SUHI) effects. However, the quantitative and localized exploration of the ISA’s influence on SUHIs in the process of urban expansion from the perspective of temporal and spatial changes is still not clear. Based on multisource remote sensing data, the SUHI effect of urban expansion is revealed by using geospatial analysis methods such as profile, difference and regression analysis. The results show the following: (1) urban expansion plays a significant role in aggravating SUHIs. Overall, the ISA and land surface temperature (LST) have obvious consistency in terms of spatial distribution patterns. However, local spatial differentiation is significant. The areas with the highest LST were not concentrated in the downtown area with the highest ISA but were scattered in the cultivated land and exposed surface areas under development in the northern part of the city. (2) In general, the ISA can explain the spatial distribution of LST well, there is an obvious positive correlation between them, and the quadratic polynomial function is the best fitting model between them. (3) The density and spatial allocation of ecological elements, such as green space and water bodies, play an important role in alleviating SUHIs. This study found that the urban center with the highest ISA coverage rate has no significant SUHI due to the reasonable allocation of green space and water bodies. The research results can provide a scientific basis for future urban planning and ecological environment construction.

Introduction

Urban heat island (UHI), a phenomenon in which the temperature in urban areas is higher than that in suburbs, which is an important representation of the effect of urbanization on local or regional microclimates and an important factor affecting the urban ecological environment and overall livability (Xie, Wang & Fu, 2011). UHIs can be categorized into atmospheric UHIs and surface UHIs. Atmospheric UHIs are measured according to the air temperature and divided into boundary layer UHIs and canopy layer UHIs; in contrast, surface UHIs are defined by measurements of the land surface temperature (LST) (Voogt & Oke, 2003; Yuan & Bauer, 2007). Surface UHIs are detected by thermal infrared remote sensing, which can characterize surface energy flow. Surface UHIs have greater temporal and spatial differentiation than atmospheric UHIs and are more sensitive to surface characteristics and human activities (Voogt, 2002). As an important parameter for characterizing surface radiation and energy exchange, LST has been widely used to study the spatial patterns of surface UHIs and their relationships with urban surface characteristics (Chen & Zhang, 2017).

Since its implementation of reform and opening up policies, China has experienced rapid urbanization, and the resulting UHIs have become increasingly serious. The existence of a UHI aggravates energy consumption and environmental pollution, reduces peoples’ quality of life and health, and causes changes in the urban ecological environment and atmospheric pattern; consequently, UHIs have become urban ecological problems worldwide (Zheng, Myint & Fan, 2014). Therefore, eliminating the UHI effect in the context of urbanization is a major challenge facing urban planning and construction and sustainable development. However, the mechanisms responsible for the UHI effect are quite complicated, as UHIs are affected by many factors, including the landscape pattern of the urban underlying surface, artificial heat sources, the urban atmospheric environment, the movement and migration of the population, the urban layout and shape, the material properties of the urban surface, and the urban weather and location (Peng et al., 2005). Among these influences, the changes in the underlying surface landscape pattern caused by urban expansion are an important driving factor of the UHI effect (Cheval & Dumitrescu, 2015; Du et al., 2016). As an important measure of the urban underlying surface, the impervious surface area (ISA) is closely related to the urban LST and is fundamental to studying surface UHI indicators (Yu et al., 2019). Impervious surfaces directly affect the vertical radiation balance by modifying the surface albedo, specific emissivity, and surface roughness, thereby intensifying the surface sensible heat flux and UHI intensity (Wei, Tan & Wang, 2014). Therefore, by quantitatively analyzing the relationship between the urban expansion of impervious surfaces and LST, the variations in the underlying surface structure and the formation and evolution of heat islands in the process of urbanization can be better understood, which is of great significance to improving the quality of the ecological environment and overall livability of urban areas.

The traditional method is to analyze the effect of urbanization on the evolution of UHIs by using temperature difference data between urban and suburban areas provided by statistical yearbooks and surface weather stations. This method has disadvantages such as the use of a single sample, low spatial accuracy, a long cycle, high cost, and an inability to accurately reflect the heterogeneity of urban land cover and thermal environment (Xie, Liu & Hu, 2016). However, the development of remote sensing technology has made up for the inadequacy of traditional research methods. Remote sensing technology has a wide detection range, can obtain data rapidly with a short time resolution and can provide a large amount of information; thus, scholars have applied this technology to study the relationship between landscape patterns and LST in the background of urban expansion (Chen & Yu, 2017; Connors, Galletti & Chow, 2013; Weng, 2001; Weng, Liu & Lu, 2007; Wu et al., 2019; Yang, Huang & Li, 2017; Yao, Xu & Zhang, 2019). The above studies shed light on how urban sprawl has contributed to the evolution of Surface UHIs. In addition, a comprehensive review of the literature shows the following: First, most of the current studies focus on the mechanism by which land cover/land use affects the thermal regulation of ecological services (Liu, Gao & Gao, 2007; Lu et al., 2020; Qian, Cui & Jie, 2006; Sarif, Rimal & Stork, 2020; Tran et al., 2017; Zhou, Huang & Cadenasso, 2011), and there is a lack of research on the quantitative relationship between key land cover components (such as ISA) and ecological factors (such as LST). Second, although some domestic scholars have studied the quantitative relationship between ISA and LST (Mao et al., 2015; Tan & Xu, 2013; Xiao et al., 2007; Yu et al., 2019), they mainly focus on megacities as their research field, but relatively little research has been done on emerging first-tier cities like Kunming. However, due to its special geographical location and climate (located in a subtropical zone but with temperate climate characteristics), Kunming has particular research value. Finally, although Song’s study (Song et al., 2014) explored the relationship between ISA change and LST in the Dianchi Lake Basin based on the difference analysis method, but only a single analysis method was used, and the specific relationship was not clear.

After nearly 20 years of development, Kunming has become an international commercial center that radiates to the whole country and faces South and Southeast Asia. With rapid urbanization and economic growth, the land used for urban construction increases greatly, which intensifies the UHI effect. Kunming, as an internationally famous tourist city and a core city in Southwest China, has a high sensitivity to the increasing effect of UHI on tourism development, ecological quality, human settlement and the urban image of Kunming (Li, Gao & Xie, 2018). How to quantitatively analyze the impact of urban expansion on the UHI effect in Kunming City from the perspective of spatio-temporal change, so as to improve the thermal environment of Kunming City, improve the living quality and health level of urban people, and enhance the environmental competitiveness of tourist cities? There has been relatively little research on this. Therefore, the main content of this article is: (1) Analyze the spatial pattern and quantitative change characteristics of ISA and LST; (2) Quantitatively study the relationship between ISA and LST; (3) Explore the impact of urban expansion on the temporal and spatial evolution of the thermal environment. In this study, landsat TM, ETM+, OLI/TIRS data and various geospatial approaches, including profile, difference and regression analysis, were used to facilitate the analysis. The research aims to provide decision-making basis for Kunming City to effectively alleviate the UHI effect, improve the somatosensory index and the comfort of the living environment, as well as for future urban planning and construction.

Materials & methods

Study area

Kunming, located within 102°10′-103°40′E, 24°23′-26°22′N, is the provincial capital of Yunnan Province, the core metropolis of the central Yunnan city group, and one of the most important central cities in western China approved by the State Council (Fig. 1). Kunming is a new first-tier city representing the whole country and a radiating example of South Asia and Southeast Asia. The main urban area of Kunming consists of five districts—Wuhua, Guandu, Xishan, Panlong and Chenggong—with 52 townships (subdistrict offices) under its jurisdiction, covering an area of 2,622 km2. In 2018, the registered population of the main urban area was approximately 2.485 million, accounting for 44.37% of the registered population of Kunming. The GDP of the main urban area reached 367.12 billion yuan, accounting for approximately 75.59% of Kunming’s GDP. The main urban area is the most economically developed and densely populated area in the city, and it is also the area with the fastest urbanization process. The city is located on the Yunnan-Guizhou Plateau. The terrain is high in the north, low in the south, uplifted in the middle, and low on the east and west sides. The overall distribution is ladder shaped from north to south. The urban area is located in Kunming Basin, with an average altitude of 1,891 m. It is surrounded by mountains on three sides and faces Dianchi Lake to the south. Kunming belongs to the low-latitude subtropical-plateau mountain monsoon climate, with an average annual temperature of 15 °C, annual precipitation of 1,035 mm, and annual temperature difference of 12–13 °C, which is the smallest range in China, without severe cold in winter or extreme heat in summer, and the four seasons are similar to spring. The UHI effect is also different from that in other cities due to the complex terrain and the particularity of the climate. Since the beginning of the 21st century, with the implementation of national strategies such as western development, new urbanization, and the Belt and Road Initiative, urban construction in the main urban area has developed rapidly; as a result, the ISA has increased significantly, and the urban atmospheric thermal environment has exhibited an increasing temperature trend. Therefore, it is imperative to explore the interrelationships of the urban expansion of Kunming’s main urban area with and its impacts on the formation and evolution of the surface UHI effect.

Figure 1 Study area.

(A) The location of the study area. (B) True color remote sensing image of the study area. Marks ① to ⑦ indicate Cuihu Lake, Dianchi Lake, Yangzong Lake, Changshui Airport or Airport New Area, Chenggong University Town, Xishan District, Guandu Lakeside, respectively.

Data descriptions and preprocessing

For the Northern Hemisphere, a narrow data selection period highlights the characteristics of the summer thermal environment, but the Yunnan-Guizhou Plateau receives considerable amounts of rain in summer, and thus, the quality of the data obtained is poor. Chen’s study (Chen & Zhang, 2017) analyzed the UHI effect in Kunming City and found that the highest UHI intensity appeared in April. To better characterize the extent of urban expansion and its thermal environment characteristics in different periods, Landsat satellite data (including ETM+, TM and Operational Land Imager (OLI) Thermal Infrared Scanner (TIRS) data) are downloaded from the United States Geological Survey (USGS) website (https://glovis.usgs.gov/). Different data collection times are employed ranging from April to May (near summer), and clear (cloudless) days of satellite transit are selected to ensure that the imaging quality is good. The detailed data parameters are shown in Table 1. In addition, to perform image preprocessing, vector data and DEM data for the main urban area are also acquired. Finally, based on ENVI software, various preprocessing operations, such as radiometric calibration, atmospheric correction, and clipping of the research area, are performed on the selected data.

Table 1 Data descriptions.

Sensor	Date	GTM	Scene ID	Spatial resolution (m)	
ETM+	30/4/2002	03:23:48	LE71290432002120SGS00	30	
TM	6/4/2008	03:24:30	LT51290432008097BKT00	30	
OLI	23/4/2014	03:34:51	LC81290432014113LGN02	30	
OLI	9/5/2020	03:34:22	LC81290432020130LGN00	30	

LST retrieval

To invert the LST in the main urban area of Kunming, this paper employs the atmospheric correction method (also called the radiative transfer equation method). The basic principle of this approach is to first subtract the estimated atmospheric influence on the surface thermal radiation from the total thermal radiation observed by the satellite sensor to obtain the surface thermal radiation intensity and then to convert this thermal radiation intensity into the LST; the expression is as follows (Sobrino, Jimenez-Munoz & Paolini, 2004):

(1) Lλ=[εB(TS)+(1−ε)L↓]τ+L↑

where: Lλ is the thermal infrared radiance detected by the satellite, ε is the surface specific emissivity, TS is the true surface temperature (K), B(Ts) is the blackbody radiance, τ is the transmittance of the atmosphere in the thermal infrared band, L↓ is the atmospheric downward radiance, and L↑ is the atmospheric upward radiance (W⋅m−2⋅sr−1⋅μm−1). Then, the blackbody radiance B(TS) at temperature T in the thermal infrared band is:

(2) B(TS)=[Lλ−L↑−τ(1−ε)L↓]/τε

In Eq. (2), TS can be obtained by Planck’s formula:

(3) TS=K2/ln(K1/B(TS)+1)

where K1=607.76 (W⋅m−2⋅sr−1⋅μm−1) and K2=1260.56K in the TM data; K1=666.09(W⋅m−2⋅sr−1⋅μm−1) and K2=1282.71K in the ETM+ data; and K1=774.89(W⋅m−2⋅sr−1⋅μm−1) and K2=1321.08K in the thermal infrared Band 10 data.

The algorithm needs only two sets of parameters: atmospheric profile parameters and surface emissivity. According to NASA’s website (http://atmcorr.gsfc.nasa.gov/), the imaging time and the central longitude and latitude can be entered to obtain the atmospheric profile parameters (L↓, L↑ and τ). The surface emissivity is divided into the emissivity of water bodies, natural surfaces and urban areas, as proposed by Qin’s study (Qin et al., 2004), and then the surface emissivity is calculated for all three types of land surfaces. To better compare and analyze the change characteristics of the surface UHI in space at different times, the LST at each time is normalized such that the LST is in the range of [0–1], and the normalized expression is used. The formula is:

(4) Ni=(Ti−tmin)/(tmax−tmin)

In Eq. (4), Ni is the normalized LST of the i-th pixel, Ti represents the LST of the i-th pixel, and tmin and tmax represent the minimum and maximum LST values, respectively.

The average (μ) and standard deviation (std) of the NDLST in these four images are then obtained, and the temperature is divided into five levels according to the density segmentation method proposed by Xu’s study (Xu et al., 2014), namely, the low-temperature (LT) zone, medium-low temperature (MLT) zone, medium-temperature (MT) zone, medium-high temperature (MHT) zone, and high-temperature (HT) zone (Table 2). In addition,the MHT and HT zones are defined as the UHI areas.

Table 2 Classification standard of temperature zone.

Temperature zone	Classification standard	
LT	NDLST < μ − std	
MLT	μ − std ≤ NDLST ≤ μ − 0.5 std	
MT	μ − 0.5 std ≤ NDLST ≤ μ + 0.5 std	
MHT	μ + 0.5 std < NDLST ≤ μ + std	
HT	NDLST > μ + std	
Note:

In this paper, μ = 0.47; std = 0.13.

ISA retrieval

Impervious surfaces in cities comprise mainly buildings and artificial ground features such as roads, parking lots, public squares and roofs, and these surfaces represent the main influencing factor of urban environmental change (Liu, Wang & Peng, 2010). The ISA, which refers to the proportion of the surface area occupied by impervious surfaces per unit area, is one of the foremost indicators used to characterize the level of urbanization development and to measure environmental quality (Arnold & Gibbons, 1996; Schueler, 1994). This paper employs the building area index (BAI) to assess the extent of urban expansion in the main urban area of Kunming. The BAI can extract artificial building surfaces such as asphalt and concrete even when the image is blurred and can effectively distinguish bare soil information (Shao et al., 2016). The expression of the BAI is as follows:

(5) BAI=((Blue−NIR))/((Blue+NIR))

where Blue and NIR are the reflectivity values in the blue and near-infrared bands of the image, respectively. In addition, because the study area contains low-reflectivity water bodies, which will affect the extraction accuracy for impervious surfaces, a modified normalized difference water index (MNDWI) (Xu, 2005) mask is used to remove water bodies. The expression of the MNDWI is as follows:

(6) MNDWI=((Green−MIR1))/((Green+MIR1))

where Green and MIR1 are the reflectivity value in the green and Mid-infrared 1 bands of the image. We first extract water from MNDWI image by setting threshold. Through continuous testing, the threshold values of water extraction in different periods are as follows: 2002, MNDWI ≥ −0.05; 2008, MNDWI ≥ 0.0; 2014, MNDWI ≥ 0.1; 2020, MNDWI ≥ 0.1. Then the non-water area is extracted based on the symmertrical difference function in Arcgis software, and finally the impervious surface is extracted from the non-water area image by BAI index.

Geometric registration is performed between the Google Earth and remote sensing images during matching periods. Based on visual interpretation, 210 training samples are randomly selected to verify the accuracy of the extracted ISA. The results show that The overall accuracy was above 86% (Except for the value of 83.64% in 2008) and the Kappa coefficient was above 0.78 (Except for the value of 0.73 in 2008), which met the recommended value by Janssen’s study (Janssen & Vanderwel, 1994). Furthermore, to facilitate an analysis of the quantitative relationship between the ISA and LST, the extracted impervious ISA is normalized. The formula is:

(7) NDISA=(BAI−BAImin)/(BAImax−BAImin)

BAImax and BAImin generally take the maximum and minimum values within a certain confidence range. The value of the confidence is mainly determined according to the actual situation of the image data of the study area. With reference to Liu’s study (Liu et al., 2009), this article uses cumulative 5% and 95% as the minimum and maximum confidence intervals. The NDISA is an important indicator that quantitatively characterizes the level of urbanization development and urban expansion in each period. The value is within the range [0–1]: the closer the value is to 1, the greater the extent of impervious surfaces; the closer the value is to 0, the more natural the surface cover. Accordingly, the NDISA are divided into 5 levels according to an equal-interval density division: low-ISA (LISA:0-0.2), medium-low ISA (MLISA:0.2-0.4), medium-ISA (MISA:0.4-0.6), medium-high ISA (MHISA:0.4-0.8) and high-ISA (HISA:0.8-1).

Geospatial measurement analysis

Profile analysis: Geographical profile analysis is widely utilized to compare, measure, and analyze the difference between a set of samples from one dimension and then to qualitatively and quantitatively summarize the results (https://blog.csdn.net). On the basis of profile analysis in the ArcGIS platform, profiles of the NDLST and NDISA values are plotted for the four phases of the image along the main expansion direction of the city (from Cuihu to Chenggong), and the value of NDISA and NDLST corresponding to the pixels passing through the profile line are extracted for quantitative analysis.

Difference analysis: The difference analysis method (also called the difference calculation technique) is used to solve the difference in a relevant or corresponding quantity. In remote sensing, the image difference analysis method is to subtract the gray value of the corresponding pixel in the multi-temporal image, and the resulting image represents the change of two temporal images (Jiang, Ma & Qin, 2005). It is the simplest and most intuitive change detection method. When applied to single band, it is called single variable image difference. When applied to multi-band, it is called multivariate image difference. To intuitively characterize the spatial heterogeneity of urban expansion and the migration and evolution of the surface UHI in the main urban area of Kunming, this paper analyzes the differences in the NDLST and NDISA among the four phases of the images.

In statistics, regression analysis is an analytical method that uses statistical principles to deal with a large number of statistical data mathematically, determine the correlation between dependent variables and some independent variables, establish a regression equation (function expression) with good correlation, and extrapolate it to predict the change of dependent variables in the future (Sheng, 2010). In order to further quantitatively analyze the relationship between ISA and LST, this paper uses ArcGIS software to reclassify NDISA at an interval of 0.01. The average of the NDLST in each interval is then calculated based on its partition statistics function. Finally, SPSS software was used to perform regression analysis on the average value of NDLST and NDISA in the interval to calculate the correlation between them.

Results

Analysis of ISA expansion characteristics

From the spatial distribution map of NDISA, the impervious surfaces of the four periods are consistent in the overall spatial distribution (Fig. 2). The areas with higher NDISA are concentrated mainly in parts of the city dominated by houses, factories, roads, and public squares (built-up areas); in contrast, areas with lower NDISA are mostly distributed on natural surfaces such as vegetation and farmland. In addition, from the perspective of the NDISA in different periods, the areas with higher ISA values have gradually expanded, indicating that Kunming’s main urban area has undergone rapid development since the turn of the 21st century, with the largest expansion and the most obvious changes occurring during 2008–2014. In 2002, the urban built-up area was distributed primarily along the Second Ring Road centered on Cuihu Lake (Mark ① in Fig. 1) and expanded outside the Third Ring Road in 2008. Chenggong District (Mark ⑤ in Fig. 1) developed rapidly in 2014. By 2020, expansion will occur less in the western and northern areas due to topographic factors, whereas expansion will continue in the other directions. The urban expansion of Kunming City as a whole presents the characteristics of “extending from the north to the south”, and the urban development has transitioned from the model of “ring Cuihu Lake” to “ring Dianchi Lake”.

Figure 2 The spatial distribution of NDISA in the main urban area of Kunming.

(A) 2002. (B) 2008. (C) 2014. (D) 2020.

To quantitatively characterize the level of urbanization development, the ISA in each period is statistically classified (Fig. 3). The results of the study show that the ISA in the main urban area of Kunming City has shown an overall increasing trend from 2002 to 2020. The area with HISA has increased year by year. From 194.13 km2 in 2002 to 344.42 km2 in 2020, the area has increased by 150.29 km2, and the proportion of impervious surfaces has increased by 6.09%. The MHISA fluctuates, with the proportion first decreasing, increasing and then decreasing. From 2002 to 2008, the proportion decreased by 1.28%; from 2008 to 2014, the proportion increased by 2.78%; from 2014 to 2020, the proportion decreased by 1.16%; the proportion overall increased over these 18 years by 0.34%. The MISA has decreased year by year. Over the past 18 years, the area has decreased by 319.99 km2, and the proportion has dropped by 13.19%. The proportion of MLISA first increased and then decreased; from 2002 to 2014, the proportion increased by 6.15% and then decreased by −8.69% from 2014 to 2020; throughout 2002–2020, the proportion showed a decreasing trend, falling by −2.54%. The LISA proportion increased first, decreased, and then increased with an increase of 1.11% from 2002 to 2008, a decrease of −7.02% from 2008 to 2014, and an increase of 15.21% from 2014 to 2020; the LISA increased by 9.3% in 18 years. Overall, the HISA, MISA and LISA changed the most over the past 18 years, while that with MHISA and MLISA changed little. In addition to the decrease in the overall proportion with MISA and MLISA, the coverage with HISA, MHISA, and LISA increased, indicating that while the level of urbanization continues to increase, the urban green environment continues to improve.

Figure 3 Statistics of ISA classification information in the main urban area of Kunming.

(A) Change trend of each ISA coverage zone; (B) area statistics of each ISA coverage zone.

Analysis of the surface UHIs temporal and spatial evolution characteristics

The NDLST can be used to reliably compare and analyze the temporal and spatial evolution characteristics of urban surface heat islands at different times; its value range is [0–1] corresponding to temperature variations from low to high. The NDLST spatial distribution map (Fig. 4) demonstrates that the overall spatial distributions of the LST and impervious surfaces are consistent. Higher NDLST values are found mainly in urban built-up areas, while the areas covered by water bodies and vegetation are predominantly cold. In addition, the spatial LST distributions in the images at different times exhibit different characteristics. In 2002, the areas with higher temperatures were distributed mainly along the Second Ring Road of the main urban area and some residential areas in the north and south, mostly with a fragmented distribution. In comparison, during 2008, the area with the most obvious temperature increase was the transition zone from the Third Ring Road (Fig. 1) to Chenggong District; the temperature change in this area is consistent with the direction of urban expansion. During this period, most expansion was distributed along a strip in the direction of urban construction and main traffic lines. From 2014 to 2020, with the exception of Dianchi Lake (Mark ② in Fig. 1), Yangzong Lake (Mark ③ in Fig. 1) and some mountainous areas with high vegetation coverage, the high-temperature area spread across the entire study area with a contiguous and concentrated distribution. We also found a significant heat island in the main city center in 2020 compared to 2014.

Figure 4 The spatial distribution of NDISA in the main urban area of Kunming.

(A) 2002. (B) 2008. (C) 2014. (D) 2020.

To better explore the evolution of the surface UHI, the classification statistics of the LST in each phase (Fig. 5) reveal the following: The LT area decreased year by year during 2002–2020 with a total decrease of 191.65 km2 and a rate of decrease of 7.18%. The MLT area first increased and then decreased, with the overall area decreasing by 78.26 km2. The MT area reached its maximum in 2014, showing a trend of first increasing and then decreasing over the past 18 years with a total decrease of 65.58 km2. The MHT area fluctuated, first decreasing and then increasing; the area overall increased by 72.76 km2. The HT area first decreased and then increased, showing an overall growth trend with a final increase of 262.38 km2 over the past 18 years. Overall, from 2002 to 2020, the MHT and HT areas exhibited an increasing trend, while the LT, MLT, and MT areas gradually decreased. These findings illustrate that the thermal environment in the main urban area of Kunming has continuously intensified. In addition, the area of the UHI declined during the period from 2002 to 2008, whereas the area rose during 2008–2020 with a rate of increase of 15.72%; hence, the period of 2008–2020 was a stage of rapid urbanization. Although the intensity of the heat island has increased, the MT area has always been dominant in various periods, which makes it difficult for the main urban area of Kunming to display an extreme climate, and thus, the urban atmospheric environment tends to be mild.

Figure 5 Statistics of LST classification information in the main urban area of Kunming.

(A) Change trend of each temperature zone; (B) Area statistics of each temperature zone.

Correlation between the ISA and LST based on profile analysis

To visually reflect the characteristics of urban expansion and the evolution of the surface UHI, the pixel points on the profile line (Fig. 6) are drawn in profile view. In 2002 (Fig. 7A), the areas with higher NDISA were concentrated mainly in the range from the city center (Cuihu) to 7.5 km along the profile. In 2008 (Fig. 7B), this extent expanded to 12 km, and in 2014 (Fig. 7C), the high-value area expanded to 25.6–27.81 km. By 2020 (Fig. 7D), the areas with both higher and lower NDISA values will expand with an alternating distribution, confirming that urban green space is constantly being established as urbanization develops. In addition, the temperature profile of the fourth-phase image shows that the NDLST of the area closest to the city center is not excessively high. The highest temperatures in 2002, 2008 and 2014 occurred at 28.6–30.1 km, 25.6-27.1 km and 13.6–15.1 km from the city center, while in 2020 they appeared at 4.5–6.0 km. Although the temperature profile does not show the typical heat island phenomenon in urban areas higher than suburban areas, the overall temperature change trend is consistent with the urban expansion trend. The profiles of the four phases all have three common features. First, NDISA is not very high (approximately 0.2–0.6) in the area of 27.8–31.8 km, while NDLST is relatively high. These areas are mainly suburban arable land, with extensive bare ground. Second, both NDLST and NDISA are relatively low in the 33.8–35.7 km area. These areas are mainly in the suburbs, with extensive green space. Finally, within 3 km of the city center, NDISA is very high (close to 1), while NDLST is not the highest. These areas are also contain a large number of water bodies and green spaces.

Figure 6 The profile line from point A to point B (black line along the direction of NDISA expansion).

(A) 2002. (B) 2008. (C) 2014. (D) 2020.

Figure 7 Surface temperature variation along a profile line in the main urban area of Kunming.

(A) 2002. (B) 2008. (C) 2014. (D) 2020.

Correlation between ISA and LST based on difference analysis

Difference analysis can visually characterize the spatiotemporal variability of urban expansion and the evolution of the surface UHI. To quantitatively characterize the urban development in Kunming, the NDISA of each period is subjected to difference analysis, and a difference image diagram is obtained between each period (Fig. 8) with difference information including the average value and standard deviation. The difference in the ISA is in the range of [−1,1]: a positive value indicates urban expansion, and a negative value indicates urban contraction; the closer the value is to the endpoint, the more obvious the change. In Fig. 8, the red color signifies increased urban construction area, whereas blue denotes the area where natural surfaces have replaced urban construction. The 2002–2008 NDISA difference image (Fig. 8A) suggests that urban construction in the main urban area was centered mainly around the Second Ring Road; the area between the Second Ring Road and the Third Ring Road is the most obvious. Chenggong University Town (Mark ⑤ in Fig. 1) also began construction. The number of facilities also increased. However, the average value of the difference image during 2002–2008 is 0.0004, the standard deviation is 0.15, indicating that urban construction was relatively slow. In contrast, the 2008–2014 NDISA difference image (Fig. 8B) shows that urban expansion occurred most evidently near the Northeast Changshui Airport (Mark ④ in Fig. 1) and the Third Ring Road to the Guandu Lakeside (Mark ⑦ in Fig. 1), followed by the rapid development of university towns. The large red area in the west represents the area burned by a forest fire that struck Xishan District (Mark ⑥ in Fig. 1) on April 16, 2014. The average value of the difference image during this period is 0.0317, and the standard deviation is 0.18, indicating that the intensity of urban construction increased during this period and that urban expansion occurred very quickly. The area with the most obvious expansion from 2014 to 2020 (Fig. 8C) is the Northeast Airport New Area (Mark ④ in Fig. 1). The average value of the difference image during this period is −0.0372, and the standard deviation is 0.16. Although there is a negative value during this period, this does not mean that the city’s level of development decreased. There are three main reasons for this negative value. First, as the level of urban development has improved, the government has invested more in ecological protection during urban planning, for example, by opening up isolation belts and large wetlands around Dianchi Lake. Second, Kunming has continuously improved its urban greening system, and the resulting increase in green space has constricted the extent of impervious surfaces. Third, land resources have been centralized and integrated, and large areas such as urban villages have been built up into residential communities, thereby reducing the plot ratio. In general, from 2002 to 2020 (Fig. 8D), the main urban area has experienced a rapid urbanization process, and the city size is constantly expanding, and the new construction land is obviously increased. The ISA of the whole region has expanded, among which Chenggong University Town and Airport New Area have the most obvious growth.

Figure 8 Image of the NDISA difference in the main urban area of Kunming.

(A) 2002–2008. (B) 2008–2014. (C) 2014–2020. (D) 2002–2008.

To show the changes in the thermal conditions at different time images more intuitively, a difference analysis is also carried out for all four phases of the NDLST images. Figure 9 demonstrates that the NDLST is consistent with the NDISA in terms of the overall spatial distribution and temporal evolution characteristics. The LST changes with the development of urban construction, but the spatial variations in the surface UHI in different periods are significantly different. From 2002 to 2008 (Fig. 9A), the range of the surface UHI temperature increase was concentrated mainly along the Second Ring Road; temperature increases were also evident in some development areas, namely, the Airport New Area in the northeast and Chenggong University Town in the south, and the surface UHIs showed a gradual shift away from the city center, while large water bodies such as Dianchi Lake and Yangzong Lake exhibited obvious cooling trends. From 2008 to 2014 (Fig. 9B), the trend was opposite to that from 2002 to 2008. The temperatures increased significantly in large water bodies, such as Dianchi Lake and Yangzong Lake Lake, and in the Xishan Mountains due to forest fires; the lakeside area of Guandu District also warmed significantly. In contrast, the main areas of expansion in the city significantly cooled because the overall temperature in 2014 was not very different, and the average LST in 2014 (0.38) after normalization was lower than that in 2008 (0.49). Compared with 2014, 2020 shows a significant temperature increase throughout the region (Fig. 9C). However, with the improvement of urban greening in Chenggong University Town and the restoration of vegetation throughout the burned forests in Xishan District, significant cooling occurred in these two regions. In general, from 2002 to 2020 (Fig. 9D), LST showed a significant warming trend with the increase of ISA. The most obvious areas of temperature increase are mainly distributed on the newly built land from the downtown of the main urban area to Chenggong University Town, showing the characteristics of “C” shape distribution along Dianchi Lake from north to south.

Figure 9 Image of the NDLST difference in the main urban area of Kunming.

(A) 2002–2008. (B) 2008–2014. (C) 2014–2020. (D) 2002–2020.

To quantitatively characterize the relationship between LST and ISA changes and further clarify the role and driving mechanism between urban expansion and the urban thermal environment, the NDLST values in each time period were reclassified by referring to the classification standard of thermal environment conditions in the zhang’s study (Zhang et al., 2007) and combining the results with the actual situation of this study. The resulting values can be divided into significantly improved (−1, −0.3], moderately improved (−0.3, −0.15], slightly improved (−0.15, −0.05], basically unchanged (−0.05, 0.05], slightly deteriorated (0.05, 0.15], moderately deteriorated (0.15, 0.3] and significantly deteriorated (0.3, 1], for a total of 7 grades. Each level corresponds to a Roman numeral followed by a digit. Then, the mean and standard deviation (STD) of NDISA within the seven grades were determined; the statistical results are as follows (Table 3).

Table 3 Mean and Std of NDISA difference at thermal variation levels.

Change	Class	2002–2008	2008–2014	2014–2020	2002–2020	
Mean	Std	Mean	Std	Mean	Std	Mean	Std	
Improve	I	0.109	0.000	0.006	0.281	0.153	0.406	0.154	0.155	
II	−0.175	0.220	−0.003	0.176	−0.360	0.629	0.585	0.243	
III	−0.098	0.214	0.031	0.153	−0.402	0.505	−0.091	0.422	
Constant	IV	−0.021	0.147	0.099	0.206	−0.146	0.404	−0.155	0.242	
Deteriorate	V	0.003	0.140	0.238	0.312	−0.037	0.186	−0.060	0.197	
VI	0.050	0.175	0.531	0.316	−0.038	0.147	0.038	0.215	
VII	0.184	0.262	0.765	0.219	0.115	0.278	0.234	0.276	

In each period, the average value of NDISA images corresponding to areas with improved thermal environmental conditions was negative (except for 2008 and 2014), indicating that a reduction in urban construction land has a certain effect on the improvement of thermal conditions. The main reason is that with the development of urbanization, the urban ecological environment is constantly improved. Urban construction land is divided or replaced by natural surfaces, such as urban green spaces, parks, street trees, wetlands and artificial lakes, which break up construction land and reduce high-temperature contiguous areas . In addition, these natural surfaces produce a “cold island effect” in local areas, neutralizing UHIs to a certain extent and improving thermal field conditions. However, in 2008 and 2014, the average value of NDISA images corresponding to an improved range of thermal conditions was positive, indicating that ISA in the improved area showed an increasing phenomenon. Similarly, 2008–2014 was the period of the fastest urbanization development in Kunming, and the difference image map for this period showed an obvious cooling phenomenon of LST within the main urban expansion area. There was a 5-year drought in Yunnan Province from 2008 to 2012, and the continuous drought led to a sharp rise in LST (Duan et al., 2015). After 2012, with the recovery of rainfall and the alleviation of drought, LST began to decline, and the thermal environment improved. However, during this period, cities were expanding and developing, so the average value of NDISA corresponding to areas with improved thermal conditions was positive. The results also show that rainfall and other meteorological changes have an important effect on the cooling of LST.

In each period, the average value of the NDISA images corresponding to thermal deterioration was positive, indicating that with an increase in NDISA, the intensity of urban construction increased, resulting in an increase in the proportion of impervious land cover, the absorption of a large amount of solar radiation, increased sensible heat flux on the land surface, and the aggravation of UHIs. In addition, from 2002 to 2008, the mean values of NDISA images corresponding to improve, constant and deteriorate thermal environmental conditions were −0.055, −0.021 and 0.079, respectively. From 2008 to 2014, the NDISA images corresponding to improve, constant and deteriorate thermal environmental conditions had values of 0.011, 0.099 and 0.511, respectively. From 2014 to 2020, the mean values of NDISA images corresponding to improve, constant and deteriorate thermal environmental conditions were −0.203, −0.146 and 0.013, respectively. The average value of NDISA difference image corresponding to the change of urban thermal environment from improved to deteriorated is from small to large in the three periods, indicating that urban expansion has a considerable effect on UHI expansion.

In general, from 2002 to 2020, the average values of NDISA images corresponding to areas with improved and deteriorated thermal environmental conditions were positive, indicating that urban construction achieved rapid development in the process of urbanization in the main urban area of Kunming City and that the overall LST also increased. However, with the increasing requirements of urban residents with respect to urban quality of life and living environment, the urban greening system also tended to improve. Although ISA increased, the thermal environment also improved. Further analysis shows that the NDISA value for the areas with medium deterioration in thermal environmental conditions from 2002 to 2020 was 0.038, which corresponds to the pixels with little change in ISA during this period. This result is mainly due to reconstruction or an increase in construction density on original construction land, such as the expansion of old cities or the reconstruction of villages within cities. In addition, the NDISA value for the areas with significant deterioration in thermal environmental conditions from 2002 to 2020 was 0.234. During this period, Kunming experienced rapid urbanization, the construction intensity significantly increased, mainly in terms of new construction land. Xie’s findings (Xie, Liu & Hu, 2016) are consistent with ours. In their study on UHIs in Wuhan, they found that the areas with relatively significant thermal deterioration were mainly distributed on original construction land, while the areas with extremely significant thermal deterioration were mainly distributed on newly added construction land.

Finally, the average STD values of NDISA images corresponding to a change in thermal status grade in four time periods—2002–2008, 2008–2014, 2014–2020 and 2002–2020—were 0.165, 0.228, 0.365 and 0.250, respectively, among which the values from 2014 to 2020 were the largest, indicating that the impact of ISA on LST in this period was not stable. In the late period of urban development, urbanization tends to focus the comfort and quality of living environments increase. The substantial increase in urban green space reduces ISA. During this period, the influence of ISA on LST decreases to some extent.

Regression analysis of ISA and LST

To compare and verify the best fitting relationship between ISA and LST, five common functions—linear, exponential, logarithmic, power and polynomial—were used to perform regression analysis on the relationship between ISA and LST, and all the fitting equations passed the significance level test of 0.01. The results show that the fitting relationship obtained with the polynomial function is better than that of other models, and its regression coefficient is significantly higher than that of other functions (e.g., the regression coefficient of the quadratic polynomial equation is 0.9291–0.9819).

The regression curve (Fig. 10) shows that the temperature in areas with higher NDISA (0.8–1) was always higher than that in areas with lower NDISA (0–0.2). Figure 2 shows that areas with higher NDISA are mainly high-density construction land, while areas with lower NDISA are mainly areas with higher vegetation coverage, indicating that construction land has a greater impact on LST than does green land. NDISA has a strong monotonic correlation with NDLST in the range of 0–0.3, while the correlation is not significant in the range of 0.7–1, indicating that in the range of low ISA coverage, with an increase in urban construction intensity, the natural surface will continue to be destroyed, so LST is more sensitive to ISA changes. In the high-value region of NDISA, there is little temperature difference within the city, and LST is not sensitive to ISA changes. Notably, NDLST reached its highest value in the range of 0.4 to 0.5. Xie’s study (Xie, Liu & Hu, 2016) proposed that NDISA values of 0.4–0.5 are the threshold range for dividing urban natural land surfaces and construction land. Figures 2 and 4 show that the area with NDISA values from 0.4–0.5 is mainly cultivated land in the north of the city and the area under development around Dianchi Lake. In addition, the regression curves of NDISA and NDLST show the “inverted U-shaped” form characteristic of polynomial functions. The NDLST value increases with increasing NDISA value, which increases rapidly at first, reaches the maximum value at 0.4–0.5, then gradually slow down , and finally tends to be stable.

Figure 10 The regression relationship between NDISA and NDLST in the main urban area of Kunming.

(A) 2002. (2) 2008. (C) 2014. (D) 2020.

SPSS software analysis revealed that Pearson’s correlation coefficient between NDISA and NDLST in four images from 2002 to 2020 was 0.818, 0.837, 0.865 and 0.901, which shows that the expansion of ISA has an intensifying effect on urban surface temperature rise and the enhancement of the UHI effect.

Discussion

Analysis of spatial relationship between ISA and LST

The results show that ISA and LST have the following spatial characteristics: On the whole, ISA and LST have consistent spatial distribution patterns. Locally, the areas with the highest LST are not concentrated in the city center of Kunming with the highest ISA, but are distributed in the cultivated land in the north of the city and the areas under development around Dianchi Lake. This finding provides an important reference basis for the future urban planning and development of Kunming City and the construction of ecological tourism destinations.

First, ISA and LST have consistent overall spatial distribution patterns, which is mainly manifested in the following aspects: (1) Urban built-up areas with higher ISA have obvious heat island effects compared with non-ISA areas such as water bodies and vegetation cover. (2) The ISA difference image map and LST difference image map were masked, and the area of ISA that greatly expanded was basically consistent with the area of LST warming. (3) Based on difference analysis, the average value of NDISA images corresponding to the areas of thermal environment deterioration in each period is positive, and the areas of significant thermal environment deterioration are mainly distributed on new construction land. Urban sprawl dominated by ISA increases the heat capacity and heat conductivity of the surface and thus intensifies the UHI effect. These findings are also comparable to some previously published results. For example, through the comparison of extracted data, Song’s study (Song et al., 2014) found that the spatial distribution of and changes in UHIs in the Dianchi Lake Basin were consistent with those of ISA. Yuan’s study (Yuan & Bauer, 2007) and Rajasekar’s study (Rajasekar & Weng, 2009) found that changes in UHI extent are consistent with the direction of urban development and expansion. Cui’s study (Cui et al., 2015) carried out research in Beijing and found that the surface temperature generally increases with urban expansion and increasing ISA.

Second, the areas with the highest LST are not concentrated in the city center of Kunming, which has the highest ISA. Profile analysis reveals that the areas with the highest NDISA values are located within 3 km of the city center, but the NDLST value of these areas is not the highest. This result may be related to the following factors. (1) The city center has a high level of urbanization, and the ecological environment is constantly being improved. The shade and evapotranspiration generated by vegetation such as green spaces, parks and street trees effectively alleviate the UHI effect (Figs. 11C and 11I). Studies have shown that green space can produce cold island effects through evapotranspiration and emissivity as well as low thermal inertia, and vegetation shade prevents direct heating of the surface by solar radiation (Li et al., 2012; Zhou, Huang & Cadenasso, 2011). This pattern is consistent with the findings of Estoque, Murayama & Myint (2017), who found that the presence of green spaces help explain why areas close to the city center do not have the highest average LST. (2) Water bodies such as artificial lakes and wetlands in the city center of Kunming play an important role in regulating the urban thermal environment (Figs. 11D and 11J). Due to the large specific heat capacity of water, the temperature increases slowly after the water body is heated, and low-temperature patches are often formed in relation to other areas. (3) Due to the high degree of urbanization in the city center of Kunming, a large number of tall buildings produce architectural shadows, which prevent direct solar radiation on the surface and reduce the sensible heat flux on the surface. As is shown in the picture (Figs. 11E and 11K), the area covered by the shadow of the high-rise building has an obvious low temperature center. We speculate that the cooling effect of high-rise buildings is extensive. Zhang’s study (Zhang & Sun, 2019) found that shading has a considerable cooling effect. Shading reduces the surface temperature of vegetation by approximately 10 °C and the temperature of the shaded part of roofs by 20–30 °C.

Figure 11 The spatial distribution map of the surface landscape and NDLST in the characteristic area of the main urban area of Kunming.

(A) The true color remote sensing image of the main urban area of Kunming in 2020. (B) The spatial distribution map of NDLST in the main urban area of Kunming in 2020. (C) to (H) are the surface landscapes of 6 characteristic areas in the main urban area. (I) to (N) correspond to (C) to (H), and (I) to (N) represent the NDLST of six characteristic regions.

Finally, the highest LST area is distributed in the cultivated land in the north of the city and the areas under development around Dianchi Lake. The cultivated land in the north of Kunming (Figs. 11F and 11L) is mainly mountainous dry land, which is far from water sources and lacks corresponding irrigation measures. Moreover, the images used in the study are mainly concentrated in April, when there is little rain, most of the land is not planted with crops, and a large surface area is exposed. Due to the destruction of original surfaces (especially natural surfaces) caused by construction in the area under development around Dianchi Lake (Figs. 11G and 11M), the surface soil has been renovated, and a large amount of the surface is exposed. Because the surface of these areas is bare and lacks the temperature regulation of water and vegetation, solar radiation directly acts on the surface, leading to a rapidly increased surface temperature. In addition, a large amount of red soil is distributed in these exposed areas. Studies have shown that red soil strongly absorbs solar radiation (Jiang et al., 2015), thus aggravating high-temperature conditions in these areas. In this study, we found that the area 27.8–31.8 km (Fig. 11H and 11N) along the urban expansion profile line and areas with regression curve values in the range of 0.4–0.5 had the highest NDLST values. Similarly, a large amount of bare land was also distributed in these areas. Future urban development should pay attention to land use in these areas and strengthen vegetation greening because UHIs in these areas are relatively easy to eliminate.

Analysis of the quantitative relationship between ISA and LST

When analyzing the quantitative relationship between ISA and LST, most scholars mainly carried out linear regression analysis and concluded that there was a linear correlation between the two parameters (Cao et al., 2011; Li et al., 2011; Yuan & Bauer, 2007). Studies by some scholars also show that the quantitative relationship between ISA and LST is not a simple linear model. They found that the exponential function is the best model (Mao et al., 2015; Tan & Xu, 2013; Wei, Tan & Wang, 2014). Our results show that the quadratic polynomial function is the best fitting model for Kunming ISA and LST. The following analysis compares the linear model (L), exponential model (E) and quadratic polynomial model (Q) to further analyze the quantitative relationship between ISA and LST.

By analyzing and comparing the scatter plots (Fig. 10) and regression equations (Table 4) of NDISA and NDLST, we found that all the models showed a good fitting relationship between ISA and LST, in which the regression coefficient (R2) of L ranged from 0.6689–0.8113, the R2 of E was between 0.6543 and 0.7915, and the R2 of Q was between 0.9291 and 0.9819. The R2 values of the three models were all greater than 0.6543, and the maximum reached 0.9819, which proved that ISA had a positive promoting effect on LST. The fitting relationships of the three models presented the trend Q>L>E, indicating that Q can best represent the quantitative relationship between ISA and LST and is the best fitting equation for the three.

Table 4 The regression models of ISA and LST.

Year	E	R2	Q	R2	
2002	y = 0.3762e0.3421x	0.6543	y = −0.3537x2 + 0.5003x + 0.3173	0.941	
2008	y = 0.4287e0.4232x	0.6819	y = −0.4621x2 + 0.6755x + 0.3515	0.9291	
2014	y = 0.3277e0.4279x	0.7364	y = −0.3377x2 + 0.5051x + 0.2699	0.9586	
2020	y = 0.5368e0.2791x	0.7915	y = −0.2954x2 + 0.4647x + 0.4858	0.9819	
Note:

All regression equations pass the significance test of 1%; y stands for LST, and x stands for ISA.

To further quantitatively characterize the impact of ISA on LST, NDLST values corresponding to different NDISA values in four periods and their changes were analyzed based on the above three models. The results show (Tables 5–7) that L cannot reflect the important rule that the changes in NDLST corresponding to high- and low-NDISA regions are different because when NDISA increases by 0.1, the increase in amplitude of NDLST in these four periods is always the same. Although E reflects that NDLST changes differently with increases in NDISA, NDLST in these four periods always shows an increasing trend when NDISA increases by 0.1, which is inconsistent with the characteristics of the regression curve (Fig. 10). Q well reflects the change in NDLST with respect to NDISA, and its change characteristics are also consistent with the regression curve. At the beginning, NDLST rises rapidly, then gradually slow down, and finally tends to be stable. Further analysis of Q shows that when NDISA ≤ 0.7, every 0.1 increase in NDISA is associated with an average increase of 0.022, 0.031, 0.024 and 0.023 in NDLST in 2002, 2008, 2014 and 2020, respectively. When NDISA > 0.7, when NDISA increases by 0.1, the NDLST corresponding to the four periods decreases on average by −0.010, −0.011, −0.007 and −0.004, respectively. That is, the difference in the average LST in high-density construction areas decreases with increasing ISA. This finding is consistent with the Li’s research (Li, Gao & Xie, 2018). They calculated the coefficient of variation of temperature (coefficient of variation = STD/mean temperature) and found that the internal overall difference in temperature in the main urban area of Kunming showed a decreasing trend against the background of an overall temperature rise.

Table 5 Leveled NDISA with its corresponding NDLST computed from the quadratic polynomial model of the four periods.

NDISA	2002	2008	2014	2020	
NDLST	Change	NDLST	Change	NDLST	Change	NDLST	Change	
0.1	0.364	–	0.414	–	0.317	–	0.529	–	
0.2	0.403	0.039	0.468	0.054	0.357	0.040	0.567	0.038	
0.3	0.436	0.032	0.513	0.044	0.391	0.034	0.599	0.032	
0.4	0.461	0.025	0.548	0.035	0.418	0.027	0.624	0.026	
0.5	0.479	0.018	0.574	0.026	0.438	0.020	0.644	0.020	
0.6	0.490	0.011	0.590	0.017	0.451	0.013	0.658	0.014	
0.7	0.494	0.004	0.598	0.007	0.458	0.007	0.666	0.008	
0.8	0.491	−0.003	0.596	−0.002	0.458	0.000	0.669	0.002	
0.9	0.481	−0.010	0.585	−0.011	0.451	−0.007	0.665	−0.004	
1	0.464	−0.017	0.565	−0.020	0.437	−0.014	0.655	−0.010	

Table 6 Leveled NDISA with its corresponding NDLST computed from the exponential function model of the four periods.

NDISA	2002	2008	2014	2020	
NDLST	Change	NDLST	Change	NDLST	Change	NDLST	Change	
0.1	0.389	–	0.447	–	0.342	–	0.552	–	
0.2	0.403	0.014	0.467	0.019	0.357	0.015	0.568	0.016	
0.3	0.417	0.014	0.487	0.020	0.373	0.016	0.584	0.016	
0.4	0.431	0.015	0.508	0.021	0.389	0.016	0.600	0.017	
0.5	0.446	0.015	0.530	0.022	0.406	0.017	0.617	0.017	
0.6	0.462	0.016	0.553	0.023	0.424	0.018	0.635	0.017	
0.7	0.478	0.016	0.577	0.024	0.442	0.019	0.653	0.018	
0.8	0.495	0.017	0.601	0.025	0.461	0.019	0.671	0.018	
0.9	0.512	0.017	0.627	0.026	0.482	0.020	0.690	0.019	
1	0.530	0.018	0.655	0.027	0.503	0.021	0.710	0.020	

Table 7 Leveled NDISA with its corresponding NDLST computed from the linear function model of the four periods.

NDISA	2002	2008	2014	2020	
NDLST	Change	NDLST	Change	NDLST	Change	NDLST	Change	
0.1	0.392	–	0.452	–	0.344	–	0.553	–	
0.2	0.407	0.014	0.473	0.021	0.361	0.016	0.570	0.017	
0.3	0.421	0.014	0.494	0.021	0.377	0.016	0.586	0.017	
0.4	0.435	0.014	0.514	0.021	0.394	0.016	0.603	0.017	
0.5	0.450	0.014	0.535	0.021	0.410	0.016	0.620	0.017	
0.6	0.464	0.014	0.556	0.021	0.426	0.016	0.636	0.017	
0.7	0.478	0.014	0.577	0.021	0.443	0.016	0.653	0.017	
0.8	0.493	0.014	0.598	0.021	0.459	0.016	0.670	0.017	
0.9	0.507	0.014	0.619	0.021	0.476	0.016	0.686	0.017	
1	0.521	0.014	0.640	0.021	0.492	0.016	0.703	0.017	

The quantitative relationship between ISA and LST is described by a polynomial fitting function for the following reasons: The suburbs with low ISA (NDISA value between 0 and 0.3) are covered with large areas of green vegetation. With an increase in ISA coverage, the vegetation in these areas is destroyed and difficult to recover in the short term, so the evapotranspiration of vegetation gradually decreases, and heat is exchanged more in the form of sensible heat, leading to a rapid rise in LST. The analysis results in Section of “Regression analysis of ISA and LST” indicate that when NDISA is between 0.4 and 0.5, this region is mainly bare surface, and a large amount of red soil is present, with a strong endothermic effect on solar radiation, leading to the highest LST in this region in the whole study area. However, the LSTs of areas with higher NDISA values (NDISA > 0.7) showed a decreasing trend, and the largest area of contiguous HISA coverage was mainly concentrated in the downtown part of the old city. In the discussion in Section of “Analysis of spatial relationship between ISA and LST”, we concluded that the widely distributed water bodies in the downtown area of the old city, the well-established green spaces and the shadows of high-rise buildings may be the main reasons for the cooling in this area. The regression curves combining NDISA and NDLST show the “inverted U-shaped” form characteristic of polynomial functions, and the regression coefficients between these variables reach 0.9291–0.9819. Therefore, we believe that the polynomial function is the best fit between the two parameters and can accurately describe the relationship between ISA and LST. The determination of this relationship has important reference value for the future urban planning and ecological environment construction of Kunming.

Some scholars’ research shows that the relationship between ISA and LST was characterized by an exponential function (Mao et al., 2015; Tan & Xu, 2013; Wei, Tan & Wang, 2014). Our research results were different from theirs, mainly because their research area was distributed in urban built-up areas and did not consider the influence of large-scale bare soil and vegetation on LST. However, in the main urban area of Kunming, exposed surfaces and vegetation are widely distributed, and the geographical location and climatic environment are unique, so ISA and LST show the characteristics of a polynomial fitting function relationship. Whether the results will be affected if the research area is restricted to built-up areas is also worth studying in the future. In addition, based on regression analysis, this paper mainly divides NDISA into intervals of 0.01 and then regresses the average values of the corresponding NDLST and NDISA data. If the whole region is sampled based on the fishing net approach, will the results also be affected? We will address this question in a future study.

Future research trends in UHI

In this study, we used remote sensing images to invert LST and ISA information and assess the dynamic quantitative impact of urban expansion characteristics on the evolution of the thermal environment. To a certain extent, this approach avoids the subjective interference caused by human factors and provides a comprehensive temporal and visual characterization of the urban surface coverage characteristics and spatial heterogeneity of thermal field conditions in Kunming over nearly 20 years. This study provides a new perspective for the study of the spatiotemporal evolution of UHIs and provides a reference for the formulation of corresponding mitigation strategies and urban construction policies. However, this paper also has some shortcomings: Due to the particularity of Kunming’s geographical location and climate environment, despite the relatively obvious characteristics of summer heat conditions. But, during this period of cloudy and rainy weather, the quality of satellite imaging was poor, so it was difficult to obtain cloudless images with relatively concentrated months. Due to the limitation of data collection, a Landsat image is used at 6-year intervals to explain the temporal and spatial changes of LST and ISA in Kunming in the past 20 years. There is a problem of ignoring subtle changes between years due to the large time span. Although the thermal infrared band of Landsat images has been widely used in the study of the UHI effect, the resolution is relatively low for the small-scale study area in the main urban area of Kunming City. Future research should be based on the particular characteristics and complexity of the research area, use higher-precision image data, shorten the time step, and combine a variety of factors and technical means to analyze the effect of urban expansion on thermal conditions to identify the driving mechanism more precisely and in greater detail. This information could further support efforts to relieve the UHI effect in Kunming and provide a scientific basis for improving the quality of urban life.

Analysis of mitigation measures for UHI

To date, scientific researchers and urban planners have analyzed the causes of UHIs and their impact on the thermal environment from different perspectives and have also begun to study measures to adapt to and mitigate UHI formation and impacts. In accordance with the relevant academic research and analysis, the authors of this article conclude that the influence of urbanization construction on the UHI effect mainly includes the following three aspects: (1) As the urbanization process accelerates, changes in the underlying surface physical and chemical properties and landscape patterns of cities occur. ISA expansion intensifies the surface sensible heat flux, concentrates the area of water and other natural vegetation surfaces and reduces the cooling effect. (2) The thermal characteristics of different surface materials are different. The ISA materials used in buildings and roads have a small specific heat capacity, rapid heat absorption, and high temperature under the sun, thus forming high-temperature patches in local cities, while roads form clear high-temperature corridors. (3) Intensive ISA is not conducive to the circulation of air, impedes the emission of heat, and intensifies the storage of heat. The above three aspects change the urban thermal landscape pattern and increase the UHI effect, which is the essential reason for changes in the urban atmospheric thermal environment. Among these factors, the change in the underlying surface caused by urban expansion, mainly ISA, is an important factor in aggravating UHIs. Therefore, this paper concludes that in future urban planning and development processes, we should focus on optimizing the proportion of urban underlying surface types. We suggest starting from the following four aspects:

First, we need to expand vegetation coverage in accordance with local conditions. The type, coverage and growth status of urban greenery determine the surface albedo and water evapotranspiration capacity, thus affecting the surface energy distribution pattern (Gallo et al., 1995; Raynolds et al., 2008; Weng & Yang, 2004). Urban greening is often discussed and is perhaps the most important strategy for alleviating UHIs. Our study found that the temperature within 3 km along the urban expansion profile line and in the area of 33.8–35.7 km is not too high because the vegetation coverage in these areas has an obvious cooling effect on the underlying surface of these locations and surrounding areas. Therefore, expanding green area coverage and reducing the floor area ratio can reduce heat storage.

Second, urban planners need to consider the spatial configuration of green space and ISA as well as the scale effect of "scattered plots" between them. In the context of global warming, the overall temperature in the main urban area of Kunming City has increased, but in the area with HISA (when NDISA is greater than 0.7), NDLST shows a trend of a slow decrease with an increase of 0.1 in NDISA. The proportion of green space and ISA is relatively reasonable because of the high urbanization level in the area with HISA coverage. Statistical analysis of the average STD values of NDISA images corresponding to a change in thermal condition grade reveals that the value for 2014–2020 is the largest because in the later period of urban development, urbanization increasingly focuses on the combination of humans and nature, and the substantial increase in urban green space reduces ISA, weakening the influence of ISA on LST. Difference analysis revealed that the average values of NDISA images corresponding to areas with improved thermal environmental conditions during 2002–2008 and 2014–2020 were negative, which was reflected in the division or replacement of urban construction land by natural surfaces such as urban green space, parks and street trees, which resulted in the fragmentation of construction land and the breaking of contiguous HT areas into smaller fragmentized areas . Therefore, the spatial allocation of green space and ISA should be considered to maximize the cooling effect of green space, expand the scale effect of well-distributed green spaces, increase the fragmentation of ISA, and thus inhibit the warming effect of ISA.

Third, it is important to consider the cooling effect of water bodies. Water plays an important role in alleviating UHI. Surface water status determines surface specific heat characteristics and is an important factor affecting regional temperature. As seen from the spatial distribution diagram of NDLST in the main urban area of Kunming City (Fig. 4), the temperature of water bodies is the lowest in the whole study area. As the mother lake of Kunming, Dianchi Lake plays an important role in influencing the local microclimate. Therefore, Kunming City should strengthen the protection of Dianchi Lake in the process of urban development and construction, increase the artificial ecological landscape of the lake, and expand the proportion of water area and wetland area.

Finally, it is important to improve the use of bare surfaces. The area 27.8–31.8 km along the urban expansion profile has the highest corresponding NDLST value, in the range of 0.4–0.5. Because there is a large area of bare ground in these areas, there is a large amount of red soil, which strongly absorbs solar radiation and the fastest temperature increase. Vegetation greening is the main method for the treatment of bare ground. At present, organic mulch technology is the main method for the treatment of bare ground in the United States. As a new type of environmental protection material, organic mulch can be described as a form of garden waste recycling. Mulching with this material aids in maintaining water and soil, increasing soil fertility, promoting plant growth, inhibiting weeds, adsorbing dust, alleviating particulate matter <2.5 µm (PM2.5), saving water and so on. Therefore, we suggest that organic mulch technology can be introduced to improve vegetation cover in exposed areas or that these areas should be converted to other land use types, such as water bodies and construction land, to reduce UHIs.

Conclusions

Based on multisource remote sensing data, this study uses geospatial analysis methods such as profile, difference and regression analysis to quantitatively localize the impact of the ISA on the SUHI phenomenon in the process of urban expansion from the perspective of spatiotemporal changes. The results show the following.

(1) From 2002 to 2020, the urban expansion of the main urban area of Kunming was obvious, with the HISA increasing by 150.29 km2, and urban development expanded from the model around Cuihu Lake to the model around Dianchi Lake. At the same time, the area of the SUHI increased by 335.14 km2, and the distribution characteristics of the SUHI transitioned from sheet-like to strip-like and then to a concentrated plane. (2) Difference analysis reveals that the average value of the normalized difference ISA (NDISA) image of the improved area is negative, while the average value of the NDISA image of the deteriorated area is positive. The average value of the NDISA difference image corresponding to the change in the urban thermal environment from improved to deteriorated is from small to large, indicating that the ISA and LST are consistent in terms of spatial and temporal changes. (3) The profile analysis shows that the highest LST area is not concentrated in the downtown area with the highest ISA but scattered in the cultivated land and developing areas in the north of the city. The large area of exposed surfaces in these areas is the main reason for the high temperatures. (4) Regression analysis shows that the ISA can explain the spatial distribution of LST well, there is an obvious positive correlation between them, and the quadratic polynomial function is the best fitting model between them. (5) This study found that the density and spatial configuration of ecological elements, such as green space and water bodies, play an important role in alleviating SUHIs. Therefore, we suggest that in future urban planning and development, the proportions of urban underlying surfaces should be optimized.

Future research should focus on the following aspects. First, the generation of SUHIs is the result of the combined action of natural and human factors, and future studies should take into account the combined effects of all warming and cooling factors. Second, a downscaling model of urban LST was established by using multisensor data sources to obtain LST data with high spatial and temporal resolution and further reveal the spatial and temporal structure rules of the SUHI, such as the spatial and temporal evolution and spatiotemporal driving model. Third, combining social and economic development status and land use data, the evaluation index system is constructed from multidimensional and multilevel data to carry out spatial measurements of a SUHI.

Supplemental Information

Supplemental Information 1 Raw dataset.

Click here for additional data file.

The authors are very grateful to the editor and anonymous reviewers for their valuable comments and helpful suggestions.

Additional Information and Declarations

Competing Interests

Author Contributions

Data Availability

The authors declare that they have no competing interests.

Xiaoliang Ma conceived and designed the experiments, performed the experiments, analyzed the data, prepared figures and/or tables, authored or reviewed drafts of the paper, and approved the final draft.

Shuangyun Peng conceived and designed the experiments, authored or reviewed drafts of the paper, and approved the final draft.

The following information was supplied regarding data availability:

The remote sensing data is available in the Supplemental File.

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
