# Peer review of "Assessing the quantitative relationships between the impervious surface area and surface heat island effect during urban expansion"

_PeerJ, doi:10.7717/peerj.11854_

## Round 0.1 · original submission · Major Revisions

Please address the comments of all the 3 reviewers and resubmit. Thanks.

Reviewer 1 ·

Basic reporting

The English need further improve

Experimental design

no comment

Validity of the findings

The authors failed to collect basic data well in the study area, like the Population, land area, socioeconomic, consumption during 2002-2020 of Kunming City, As a result, the conclusion reached in the discussion is not supported by strong evidence.

Additional comments

Title: Assessing the quantitative relationships between the impervious surface area and heat island effect during urban expansion: a case study in Kunming, China
ID: #55463
Conclusion: Reject
Comments for authors:
Abstract:
(1) Lack of presentation of key quantitative results, some important data need present.
Introduction:
(2) L37: An UHI not A UHI
(3) L71: Statistical yearbook no references
(4) Introduction should point out the situation of urban expansion during 2002-2020 years, and emphasize the importance of studying the UHI to Kunming City.
Materials & Methods
(5) Please added the area, population, and some basic information about Kunming City.
Results
(6) L240-247 should move to section of Materials & Methods
(7) Please add a distribution figure of Kunming City and clearly note the place name mentioned in your text, such as: First, second, and third Ring, Dianchi Lake, Yangzonghai, guandu, etc. Because readers not familiar with your study area.
(8) L321, where is the Fig A1?
(9) L460, where is [17]?
(10) Please add a regression figure to present the regression result.
Discussion
(11) L445-447, where is the reference or evidence for mentioned viewpoint?
(12) L481-495,The author admits that linear regression is flawed, why not use
nonlinear or multiple comparisons in current text? Present data completely satisfying the requirement of the method for nonlinear regression.
(13) The authors failed to collect basic data well in the study area, like the Population, land area, socioeconomic, consumption during 2002-2020 of Kunming City, As a result, the conclusion reached in the discussion is not supported by strong evidence. Such as L437-439, L457-459, L451-453, L518-530.
(14) L531-586, you mentioned 3 measures are irrelevant to your results in text.
Conclusion
(15) Rephrase
Overall, I think the manuscript should be rejected.

Reviewer 2 ·

Basic reporting

Please refer to the general comments section.

Experimental design

Please refer to the general comments section.

Validity of the findings

Please refer to the general comments section.

Additional comments

The paper studies the relationship between LST and ISA in Kunming, China. The findings of the paper can be beneficial for the researchers in the field. However, the followings should be addressed.

There is a jump from the first to the second sentence of the abstract, without providing any information about the importance of doing this analysis on Kunming, China as a case study.
The abstract lacks cohesion and needs to be revised.
It is commonly known that there is a strong relationship between ISA and LST. So, the authors should specify how the findings of this paper will help urban planners and researchers.
The innovation of the research is not clearly specified.
Line 42: the sensitivity of LST to surface characteristics cannot be the reason for the current trend of studying UHI.
Line 243: 80% accuracy is not high. How does this satisfy the requirements?
Line 396: what does good represent here?
Line 419: “The heat island effect is related to the size of the city. The larger the scale is, the more obvious the heat island effect (that is, the greater the temperature difference between urban and rural areas)”. The statement needs reference.
Line 424: “that the urban atmospheric environment tends to be mild”. How can the results prove this statement while the research was done using LST?
Line 437: “Therefore, the background climate, the geographic location, the surrounding topography and even the collection times and seasons of different satellite data should be considered for different research areas.” How does this statement help the findings of this paper?
Line 481: “To analyze the quantitative relationships between impervious surfaces and LST, this article employs linear regression analysis. However, many scholars’ studies have shown that the quantitative relationships between impervious surfaces and LST are not simply linear.” If the literature already showed that the relationship between ISA and LST is not linear, why linear regression was used in this study?
Line 494: needs revision.
In the discussion part, the author mainly analyzed other works, while it is expected to justify the results of the current study with the results of other studies.
Figures and tables:
It is not clear what the right-hand section of Figure 1 shows.
The quality of the graphs should be increased.
The size of the figure 3 can be decreased. It would be easier for the reader to follow the trend if it was a linear graph.
The paper needs to be majorly revised in terms of grammar. Some of the examples that needs to be modified are listed below:
Line 15: Conjunctions were not used correctly.
Line 19: word “analysis” was used three times in a row.
Line 16: the sentence is very long and can be difficult for the reader to follow.
Line 61: “effect(Cheval & Dumitrescu 2015; Du et al. 2016)”. A space is required between the word “effect” and parenthesis. The same problem can be seen in line 63, 101, 192, …
Line 34: paragraph started without indentation while in line 48 the paragraph started with indentation.
Line 39: “boundary-layer heat islands and canopy-layer heat islands”; Unnecessary repetitions of words.
Line 72: “urban development, etc.”
Line 73: needs revision.
Line 71: a very long sentence.
Line 78: “the traditional methods cannot effective reflect”. grammatical error
Line 105: “this paper selects an impervious surface index”. Select might not be a suitable verb here.
Line 105: Very long sentence. Paper needs to be majorly revised in terms of the structure of sentences.
Line 114: “UHI effect for Kunming”. on Kunming?
Line 263: “revealing that the ISA in the main urban area of Kunming has been characterized by an overall increasing trend from 2002 to 2020”. needs revision.
Line 421: the sentence started with “However” while there is no contrast between the previous sentence and this sentence.
Line 423: conjunction was not used correctly.
Line 419: a very long sentence.

Reviewer 3 ·

Basic reporting

Please see 'General comments for the author'.

Experimental design

Please see 'General comments for the author'.

Validity of the findings

Please see 'General comments for the author'.

Additional comments

This study proposes the quantitative relationship between the impervious surface area and heat island effect during urban expansion in Kunming, China. While it has potentials to be published, there are some issues which need to be addressed before proceeding further.

• Overall, the study lacks innovation.
• English improvement is required as there are a lot of grammatical errors.
• Lines 200 & 206: LC indices value ranges for different LC types should be indicated. See the paper below as an example:
“Remote sensing image-based analysis of the relationship between urban heat island and land use/cover changes”
• Table 3: It is highly recommended to add the scatterplots of random samples to show the comparison clearly.
• Figure 5: Using only one Landsat image every 6 years is not enough to explain the temporal variation of the SUHI over two decades. Now, Google Earth Engine platform enables RS researchers to process hundreds of images during long time periods to increase the accuracy of obtained results.
• Line 340: Correlation between ISA and LST based on difference analysis
There are several places’ names in this part with unknown locations, which can make readers confused, like Chenggong University City, Xishan District, Airport New Area, Dianchi Lake, Yangzonghay ….
All places should be illustrated on a map. You can show them on the map used to show the study area (Figure 1)
• Figure 1: The size of the border of Yunnan Province should be increased.
The map on the right side of this Fig and Table 1 are giving the same information. You do not need to have both.
• Some statements need to be referenced. For examples:
‘Geospatial measurement analysis’: Profile analysis (Line 214), Difference analysis (Line 221), …
‘Discussion’: Line 419
• There is no space between the words and in-text references. Fx:
Line 37: … overall livability of urban areas(Xie et al. 2011).
Line 43: … surface characteristics and human activities(Voogt 2002),…
Line 59: … the urban weather and location(Peng et al. 2005).
Spaces should be added.
• Line 417: The Discussion part should mainly discuss the results obtained in the study. The main purpose is to compare the results of a study with the literature and the implications of similarities and differences. But, in this study, the authors separated the discussion part from the results, and mainly reviewed the other similar works.

---

## Round 0.2 · Minor Revisions

Thanks for the revisions. Please go through the suggestions suggested by the two reviewers and resubmit. I agree with the minor revisions suggested by one of the reviewers, and also with the edits suggested by the other reviewer. Thanks.

Reviewer 2 ·

Basic reporting

Please refer to "general comments for the author".

Experimental design

Please refer to "general comments for the author".

Validity of the findings

Please refer to "general comments for the author".

Additional comments

The authors did make modifications.
As the abstract and conclusion parts are too long, it is highly recommended to revise these two parts.
Keywords are missing.

Reviewer 3 ·

Basic reporting

-

Experimental design

-

Validity of the findings

-

Additional comments

The written English has been improved throughout the manuscript in the revised version.
As UHI and SUHI have different characteristics and in this manuscript LST has been explored, it is suggested to mention surface heat island rather heat island in the title.
Summarising the abstract is highly recommended, and then it would have the potential to be published in the journal.

---

## Round 0.3 · accepted · Accept

Thank you for making all the revisions. This paper can be accepted for publication.